

# Identifying Dominant Parameters Across Space and Time at Multiple Scales in a Distributed Model Using a Two-Step Deep Learning-Assisted Time-Varying Spatial Sensitivity Analysis

Jing Yang[1,2], Jiangjiang Zhang[3,4], Tian Jiao[1], Yonghua Zhao[2], Manya Luo[2], Lei Wu[5], Ming Ye[6], Jinxi Song[1]

[1]Key Laboratory of Environmental Simulation and Ecological Health in the Yellow River Basin, College of Urban and Environmental Sciences, Northwest University, Xi'an 710127, China

[2]School of Land Engineering, Key laboratory of Degraded and Unused Land Consolidation Engineering of the Ministry of Natural Resources, Shaanxi Key Laboratory of Land Consolidation, and Shaanxi Province Land Consolidation Engineering Technology Research Center, Chang'an University, Xi'an 710054, China

[3]State Key Laboratory of Water Disaster Prevention, Hohai University, Nanjing 210098, China

[4]Yangtze Institute for Conservation and Development, Hohai University, Nanjing 210098, China

[5]College of Water Resources and Architectural Engineering, Northwest A&F University, Yangling 712100, China

[6]Department of Earth, Ocean, and Atmosphere Science and Department of Scientific Computing, Florida State University, Tallahassee 32306, USA

*Correspondence to* J. Tian (tianjiao@nwu.edu.cn) and Y. Zhao (yonghuaz@chd.edu.cn)

**Abstract.** Distributed models require parameter sensitivity analyses that capture both spatial heterogeneity and temporal variability, yet most existing approaches collapse one of these dimensions. We present a two-step, deep learning-assisted, time-varying spatial sensitivity analysis (SSA) that identifies dominant parameters across space and time. Using SWAT for runoff simulation of the Jinghe River Basin, we first apply the Morris method with a spatially lumped strategy to screen influential parameters and then perform SSA using a deep learning-assisted Sobol' method for quantitative evaluation. A key innovation lies in the systematic sensitivity evaluation with parameters represented and analysed at both subbasin and hydrologic response unit (HRU) scales, enabling explicit treatment of distributed parameters at their native spatial resolutions. To reduce computational burden, two multilayer perceptron surrogates were trained for 195 subbasin and 2,559 HRU parameters, respectively, allowing efficient time-varying SSA of NSE-based Sobol' indices over 3- and 24-month rolling windows during 1971-1986. Results reveal structured, scale-dependent controls: spatially, sensitivity hotspots are coherent between scales but become more localized at the HRU level, reflecting heterogeneity in land use, soils, and topography; temporally, sensitivities fluctuate with runoff in the 3-month window, while event-scale variations are smoothed in the 24-month window, yielding more persistent patterns governed by storage and routing processes. The proposed framework provides a computationally efficient and unified approach for identifying scale-dependent sensitivity hotspots and hot moments, thereby supporting targeted calibration and enhancing the interpretability and predictive robustness of distributed models under nonstationary conditions.



## 1 Introduction

Runoff simulation underpins hydrologic prediction and decision-making, supporting applications ranging from short-term
flood warnings (Su et al., 2024; Kilicarslan and Temimi, 2024; Ou et al., 2025) and water allocation to ecosystem
management (Dalcin and Fernandes Marques, 2020; Uday Kumar and Jayakumar, 2021; Bai et al., 2025a;) to climate-
adaptation planning (Hou et al., 2022; Zhou et al., 2023; Chitsaz et al., 2025). The reliability of these simulations depends
critically on how hydrologic models trransform rainfall into runoff. Broadly, such models fall into two classes according to
their degree of spatial discretization: lumped and distributed (including semi-distributed variants) formulations (Fatichi et al.,
2016; Pelletier and Andréassian, 2022; Bhanja et al., 2023). Lumped models treat the entire catchment as a single or a few
effective units, providing computational efficiency and requiring fewer parameters, but at the cost of neglecting spatial
heterogeneity in land surface characteristics. Distributed models, by contrast, explicitly account for spatial variations in soil,
land cover, and topography by partitioning the basin into Hydrological Response Units (HRUs) or grid cells with spatially
varying parameters. This enhanced realism improves process representation and transferability across basins but also
introduces high-dimensional parameter spaces and substantial computational burden. Because not all parameters exert
influence everywhere or at all times, a key scientific challenge arises: *how does parameter effects vary across space and
time, and how do spatial and temporal scales reshape the sensitivity structure of distributed models*?

Sensitivity analysis (SA) provides a systematic framework to quantify how variations in input parameters affects model
output. In distributed runoff modeling, SA helps identify dominant parameters, guide calibration, and enhance process
understanding. Conventional SA approaches can be grouped into spatially lumped and spatially distributed methods.
Spatially lumped SA assumes uniform parameters or uniform scaling factors across the basin, offering computational
efficiency but masking spatial heterogeneity (Zhang et al., 2013; Khorashadi Zadeh et al., 2017; Koo et al., 2020; Acero
Triana et al., 2025). In contrast, spatially distributed SA (SSA) quantifies sensitivity at each spatial unit, revealing spatial
patterns of hydrologic control and identifying sensitivity hotspots where certain parameters dominate (Tang et al., 2007; van
Werkhoven et al., 2008b; Rouzies et al., 2023). SSA results often highlight the linkage between physical processes and
catchment characteristics, e.g., infiltration parameters dominating in permeable uplands and channel-loss parameters in arid
lowlands.

However, parameter influence is inherently dynamic. Studies have shown that sensitivities shift over time as hydrologic
states, climate drivers, and antecedent conditions change (Wagener et al., 2003; Sieber and Uhlenbrook, 2005; van
Werkhoven et al., 2008a; Pfannerstill et al., 2015). Building on this insight, Herman et al. (2013b) introduced the "time-
varying SA" concept, employing a moving-window approach to visualize how dominant processes shift seasonally.
Subsequent developments refined this concept. For example, Massmann et al. (2014) proposed a wavelet-based approach to
visualize the evolution of parameter sensitivity across multiple temporal scales. Pianosi and Wagener (2016) used time-
varying SA to disentangle the temporal evolution of parameter and data uncertainties. Xie et al. (2017) extended it to assess
hydrologic and sediment parameters across varying timescales. Beyond runoff modeling, Quinn et al. (2019) extended the



time-varying SA framework to reservoir operation systems, showing how sensitivities evolve under varying inflow and storage regimes. More recently, Yang et al. (2024) extended the framework to a biogeochemical model of trichloroethylene degradation, demonstrating dynamic shifts in sensitivities of flow and reactive transport parameters. Collectively, these studies confirm that parameter influence fluctuates with climatic drivers, antecedent moisture, and event magnitude. This is

70 vital for adaptive calibration, uncertainty quantification, and process understanding.

A major conceptual advance was made by Herman et al. (2013a), who pioneered the "from maps to movies" concept by extending time-varying SA to spatially distributed hydrologic models, i.e., the time-varying SSA. Their framework computes parameter sensitivities for each spatial unit within sequential moving windows, transforming static sensitivity "maps" into dynamic "movies" that reveal how dominant controls evolve through both space and time. The framework captures "hot

moments" (periods of heightened sensitivity) that complement spatial "hotspots," providing an intuitive visualization of how hydrologic controls vary under changing conditions. Building upon this idea, Wu et al. (2022) implemented time-varying SSA within a grid-based mHM-Nitrate model to identify key parameters governing NO3-N dynamics. By employing a one-year moving window at each grid cell, they effectively captured spatiotemporal shifts in parameter importance across hydrological regimes. These applications underscore the capability of time-varying SSA to reveal process dynamics that

conventional static analyses overlook.

Despite these advances, two challenges persist. First, the computational demand of distributed models is enormous. Performing SA across numerous spatial units multiplies the parameter space, often by orders of magnitude, rendering comprehensive spatiotemporal exploration infeasible. Consequently, our understanding of spatiotemporal sensitivity patterns in distributed models remains fragmented. Second, sensitivity is inherently scale dependent. Sensitivity patterns vary with

85 both spatial resolution (e.g., subbasin vs. HRU) and temporal aggregation (e.g., daily, monthly, seasonal). Changes in spatial discretization alter hydrologic connectivity and parameter dominance (Saint-Geours et al., 2014; Guse et al., 2016; Acero Triana et al., 2025), while varying temporal windows modulates the relative influence of fast- and slow-response processes (Massmann et al., 2014). Yet few studies have systematically examined how spatial and temporal scales jointly reshape sensitivity structure and whether dominant controls persist across scales.

Here, we fill these knowledge gaps by developing a two-step, time-varying SSA framework that integrates deep-learning surrogates with the Soil and Water Assessment Tool (SWAT). The workflow proceeds in three steps: (1) initial Morris method to screen influential parameters related to runoff simulation, (2) training of subbasin- and HRU-scale multilayer perceptron (MLP) surrogates to approximate SWAT's runoff series, and (3) computation of time-varying Sobol' indices at different spatial scales using moving windows of different lengths. MLP surrogates provide a computationally efficient

mapping between high-dimensional parameter spaces and model outputs (Furong and Hossain, 2023; Jahanshahi et al., 2025; Yu et al., 2024), reducing computational burden and enabling time-varying SSA that would otherwise be prohibitive. Analyses are conducted for the Jinghe River Basin, a typical semi-arid watershed in northwest China characterized by strong topographic gradients and marked hydrologic seasonality. Two spatial scales (subbasin and HRU) and two temporal scales (3- and 24-month window lengths) are used to examine how sensitivity patterns reorganize across spatiotemporal scales.





Specifically, the study aims to: (1) develop a computationally efficient framework for spatiotemporal SA in distributed hydrologic models; (2) identify dominant parameters and their spatial and temporal dynamics controlling runoff generation; and (3) quantify how spatiotemporal scales reshape sensitivity hotspots and hot moments. By achieving these goals, we aim to advance understanding of how distributed parameters jointly control hydrologic behavior across spatial and temporal scales, thereby enhancing both the diagnostic and predictive capability of physically based models.

## 2 Methodology

### 2.1 Study Area

The Jinghe River Basin, located in the central Loess Plateau of China, drains an area of $4.4 \times 10^4$ km$^2$ between 106°14'E to 108°42'E and 34°46'N to 37°19'N (Figure 1a). The mainstem is ~455 km long and discharges to the Wei River (the largest tributary of the Yellow River). The basin typifies an agriculture-dominated region situated in the transition zone between semi-humid and semi-arid climates, characterized by severe soil erosion (Bai et al., 2022; Bai et al., 2025b), fragile ecosystems (Xu et al., 2022), and high hydrological variability (Chang et al., 2016; Liu et al., 2024). The regional climate is strongly influenced by the East Asian monsoon, producing pronounced seasonality in rainfall. Daily records from four meteorological stations (Figure 1c) during 1962 - 1986 indicate a mean annual temperature of 8.9 °C and a mean annual precipitation of 654 mm, ranging from 381 mm to 857 mm. Nearly 55% of rainfall occurs between July and September. Mean annual evapotranspiration, estimated by using the Penman-Monteith method for the same period, is 972 mm. Evapotranspiration exceeds precipitation from October through June of next year when rainfall is scarce (Figure 1b).

Topography declines from northwest to southeast, from 2,884 m to 219 m above sea level (Figure 1c). Consistent with this gradient, flow routes from the uplands in the northwest to the Wei River confluence in the southeast. Long-term (1956 - 2016) mean annual runoff is estimated at $15.66 \times 10^8$ m$^3$ at the Zhangjiashan (ZJS) gauge station (Figure 1c), with ~55% of the flow occurring from July to October (YRCC, 2023). Land cover is dominated by cropland and pasture (82.2%; Figure S1a). Calcaric cambisols are the prevailing soil group, covering 67.9% of the basin (Figure S1b). The interplay of intensive cultivation, highly erodible loess, and uneven precipitation creates persistent challenges for ecological sustainability and robust hydrologic modeling, motivating sensitivity analyses that resolve controls across spatial and temporal scales.





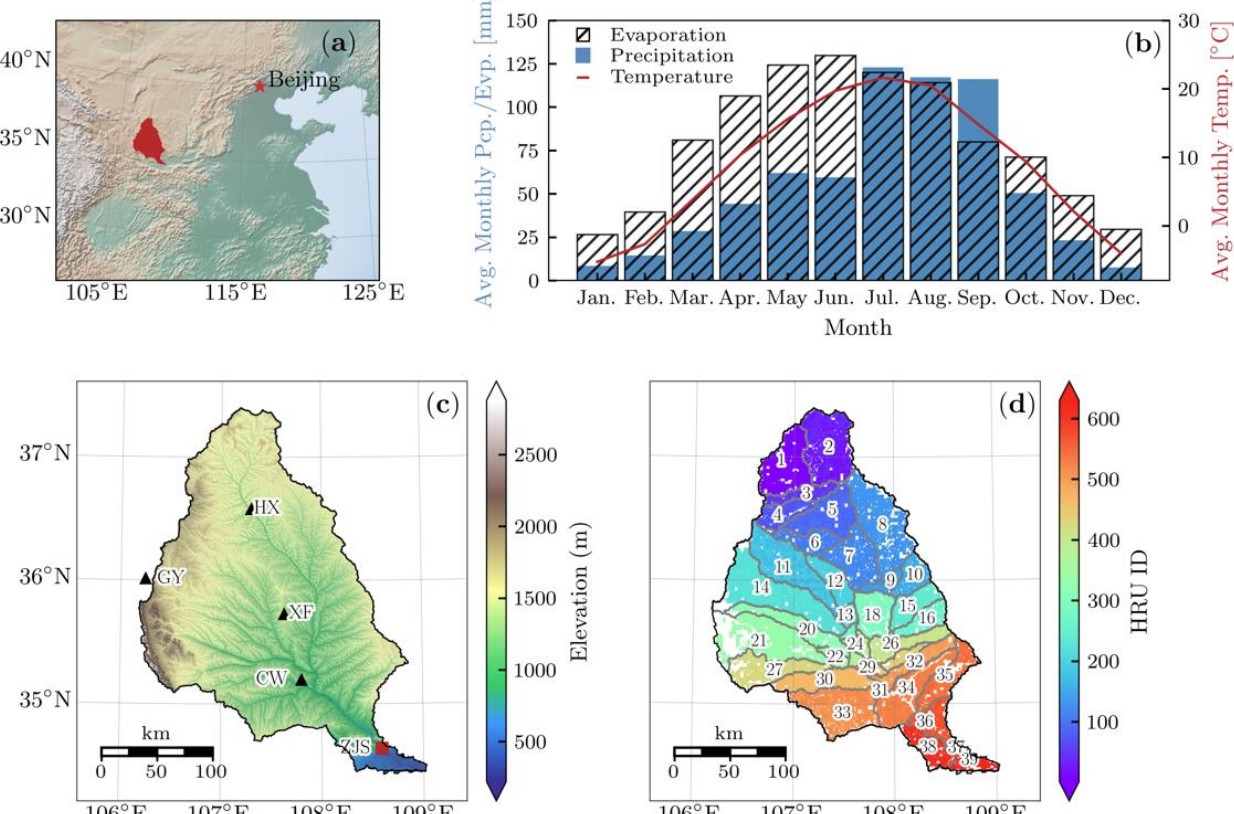

**Figure 1: (a) Location of the Jinghe River Basin. (b) Monthly mean climate characteristics during 1962 - 1986, including temperature (red line), precipitation (blue bars), and evaporation (black hatches). (c) Basin topography based on a digital elevation model (DEM) obtained from the Geospatial Data Cloud (http://www.gscloud.cn/, accessed September 25, 2025). Black triangles denote four meteorological stations Huanxian (HX), Guyuan (GY), Xifeng (XF), and Changwu (CW), and the red square marks the Zhangjiashan (ZJS) gauge station. (d) Delineation of 39 subbasins and 630 HRUs. Numbers label subbasin IDs; colors indicate HRU IDs (some subbasin labels omitted to avoid overlap).**

## 2.2 Morris and Sobol' Methods for Global Sensitivity Analysis

The Morris method, also known as the Elementary Effects (EE) method (Morris, 1991), provides an efficient, qualitative screening of influential parameters in high-dimensional models (Gao and Bryan, 2016; Wu et al., 2022). Let the model output be $y(\mathbf{x})$, with $\mathbf{x} = (x_1, x_2, …, x_k)$ the vector of $k$ uncertainty parameters. Each parameter is mapped to the unit interval [0, 1] (e.g., by using its cumulative distribution), which is then discretized into $p$ levels, yielding a perturbation step size of $\delta$ = 1 / ($p$ - 1). For a randomly selected base point on this $p$-level hypercube grid, the EE of $x_i$ (denoted as $EE_i$) is computed by a one-at-a-time perturbation of size $\delta$ with $(x_i + \delta) \in [0, 1]$, while holding other parameters fixed (Yang and Ye, 2022):

$$EE_i = \frac{y(x_1,…,x_i+\delta,…,x_k) - y(x_1,…,x_i,…,x_k)}{\delta}. \tag{1}$$

Repeating this perturbation from multiple random base points yields multiple $EE_i$ values, from which three statistics are derived: the mean ($\mu_i$) of $EE_i$, the standard deviation ($\sigma_i$) of $EE_i$, as well as the mean ($\mu_i^*$) of $|EE_i|$. Following Campolongo et





al. (2007), $\mu_i^*$ is preferred for identifying influential parameters because it avoids cancellation of effects with opposite signs. The $\sigma_i$, reflects the dispersion of elementary effects and is used to detect parameters whose influence on the output is nonlinear or involves interactions with other parameters (Yang and Ye, 2022). Parameters with low $\mu_i^*$ and $\sigma_i$ are considered non-influential and can be excluded from further analysis (Sreedevi and Eldho, 2019; Dai et al., 2024).

Equation (1) suggests that estimating $r$ EEs requires a total of $2rk$ model simulations, since each EE needs a base and a perturbed evaluation. Morris (1991) proposed a trajectory-based design to reduce this cost. Starting from a random base point, the design applies $k$ successive one-at-a-time perturbations, each modifying a single parameter, to construct a trajectory of $(k + 1)$ points. Repeating this process $r$ times yields $r$ trajectories and a total of $r \times (k + 1)$ model evaluations. This design substantially lowers the computational cost compared with the naive scheme, but its results remain qualitative.

To move from qualitative screening to quantitative attribution, we use the Sobol' method. Sobol' method (Sobol', 2001; Saltelli et al., 2007) addresses this need by decomposing the total variance of model output into contributions from individual parameters and their interactions. For a model output $y(\mathbf{x})$, Sobol' sensitivity indices give a rigorous measure of parameter importance. The first-order sensitivity index $S_i$ measures the main effect of a parameter $x_i$ and the total-effect sensitivity index $S_{Ti}$ quantifies both the main effect and all interactions (Saltelli et al., 2007):

$$S_i = \frac{V_{\mathbf{x}_i}\big[E_{\mathbf{x}_{\sim i}}(y|x_i)\big]}{\mathrm{V}(y)}, \tag{2}$$

$$S_{Ti} = \frac{E_{\mathbf{x}_{\sim i}}\big[V_{\mathbf{x}_i}(y|x_{\sim i})\big]}{\mathrm{V}(y)} = 1 - \frac{V_{\mathbf{x}_{\sim i}}\big[E_{\mathbf{x}_i}(y|x_{\sim i})\big]}{\mathrm{V}(y)}, \tag{3}$$

where $E$ depict the mean and $V$ represent the variance; $x_{\sim i}$ represents all parameters except $x_i$. According to Saltelli et al. (2007), $S_i$ supports "*Factors Prioritization*" by ranking parameters according to their direct importance, while $S_{Ti}$ supports "*Factors Fixing*" by identifying parameters that can be fixed without significantly affecting model output.

In practice, efficient estimation of Sobol' indices follows the Saltelli sampling scheme (Saltelli et al., 2007). The procedure begins with the generation of two independent sampling matrices, $\mathbf{A}$ and $\mathbf{B}$, both of size $N \times k$, representing $N$ sample size and $k$ input parameters. Then for each parameter, a hybrid matrix $\mathbf{C}_i$ is formed by replacing the $i$-th column of matrix $\mathbf{B}$ with the corresponding column from $\mathbf{A}$, while retaining all other columns of $\mathbf{B}$. The first- and total-order sensitivity indices, $S_i$ and $S_{Ti}$ of the $k$ parameters, are then estimated using $2 \times N$ model simulations from $\mathbf{A}$ and $\mathbf{B}$, plus $k \times N$ model simulations from

the hybrid matrices $\mathbf{C}_i$, resulting in a total cost of $N \times (k + 2)$ model runs (Yang et al., 2024). Although this remains computationally demanding, it provides a robust quantitative basis for parameter importance. This estimator is widely adopted in modern SA toolkits, including SALib (Herman and Usher, 2017) and the SAFE Toolbox (Pianosi et al., 2015).

## 2.3 SWAT Model Description and Configuration

SWAT is a semi-distributed hydrological model that simulates water, nutrient, and pesticide transport at the catchment scale
(Zhang et al., 2014; Aloui et al., 2023). In SWAT, a basin is typically divided into multiple subbasins, which are spatially referenced units. Each subbasin is further subdivided into HRUs, characterized by distinct combinations of land use, soil



type, and slope category (Golmohammadi et al., 2017; Jeong et al., 2024). HRUs are not necessarily contiguous or spatially explicit within the model domain, but each is assumed to be homogeneous with respect to land surface characteristics that govern hydrologic processes. They constitute the smallest computational units in SWAT (G. Arnold et al., 2012). All
hydrological processes are driven by the soil-water balance, which is governed by (Neitsch et al., 2011):

$$SW_t = SW_0 + \sum_{i=1}^{t}(R_{day,i} - Q_{surf,i} - E_{a,i} - W_{seep,i} - Q_{gw,i}), \tag{4}$$

where $SW_t$ and $SW_0$ is the final and initial soil water content (mm $H_2O$), $R_{day,i}$ is daily precipitation (mm $H_2O$) on day $i$, $Q_{surf,i}$ is surface runoff (mm $H_2O$) on day $i$, $E_{a,i}$ is evapotranspiration (mm) on day $i$, $W_{seep,i}$ is water percolation from the soil profile into the vadose zone (mm) on day $i$, and $Q_{gw,i}$ is groundwater return flow (mm $H_2O$) on day $i$.

Variety datasets were prepared to configure the SWAT model for the Jinghe River Basin and they are detailed below:

(1) The DEM was sourced from the ASTER GDEM V2 (30 m resolution), obtained from the Geospatial Data Cloud (http://www.gscloud.cn/, accessed September 25, 2025). The DEM was utilized to delineate watershed boundaries, extract river networks and subbasins, and determine slope classes.

(2) Soil map was obtained from the Harmonized World Soil Database (HWSD, 1 km resolution), available through the
National Cryosphere Desert Data Center (https://www.ncdc.ac.cn/portal/, accessed September 25, 2025).

(3) The land use map for the year 1980 (30 m resolution) was provided by the Geographical Information Monitoring Cloud Platform (http://www.dsac.cn/, accessed September 25, 2025). The 1980 dataset was chosen to reduce the influence of human impacts, such as major soil and water conservation programs (e.g., the "Grain-for-Green" project) since 1980s (Chen et al., 2007).

(4) Daily runoff records from the ZJS gauge station (Figure 1c), covering the period of 1962 - 1986, were retrieved from National Earth System Science Data Center (http://www.geodata.cn/, accessed September 25, 2025) and the Hydrological Yearbook published by the Ministry of Water Resources of China.

(5) Daily meteorological data for precipitation, air temperature, relative humidity, and wind speed were collected from four meteorological stations (Figure 1c), via the Shaanxi Provincial Meteorological Bureau (http://sn.cma.gov.cn/,
accessed September 25, 2025) for 1962 - 1986. Solar radiation was not directly measured and was estimated using SWAT's weather generator (WGEN).

To reduce computational cost while preserving basin geometry, we resampled the DEM to 150 m after trial-and-error tests verified that subbasin boundaries and drainage structure remained consistent with the original 30 m resolution. Besides, the land use was resampled to 3,000 m, which had negligible impact on HRU definition. Using an 80,000 ha drainage-area
threshold, the basin was delineated into 39 subbasins. These subbasins was subdivided into 630 HRUs using the *multiple HRUs* option in SWAT by overlaying soil, land use, and slope maps, with thresholds of 10% for soil, 15% for land use, and 10% for slope (Figure 1d). Note that there are blank areas during the HRU generation process (Figure 1d), which correspond to areas filtered out by these thresholds or to small spatial mismatches among layers. The full subbasin-HRU crosswalk is provided in Table S1.



The model was simulated at a monthly time step over 1962 - 1986, consistent with the available meteorological data. The first 10 years (1962 - 1971) were used as a warm-up period, and the subsequent 15 years (1972 - 1986) were analyzed for sensitivity and evaluation. In this study, rainfall-runoff processes were simulated using the Soil Conservation Service (SCS) Curve Number method and channel flow routing was represented by the variable storage (variable travel time) method (Neitsch et al., 2011). It is also worth noting that no model calibration was performed in this study. The study's objective was to quantify how parameter uncertainties control runoff across space, time, and scale, rather than to optimize predictive skill. All parameters were kept at default values to avoid calibration-induced compensations and to maintain diagnostic integrity for the subsequent sensitivity analysis.

**2.4 Parameter Screening Using the Morris Method with a Spatially Lumped Strategy**

The SWAT model contains a large number of parameters, potentially hundreds, depending on spatial discretization and the processes represented. Conducting spatiotemporal SA across the full parameter space is computationally infeasible, making preliminary screening necessary to identify the most influential parameters governing runoff dynamics. Guided by prior studies and domain expertise, we selected 33 parameters commonly recognized as important for runoff generation (Table S2). All parameters were assumed to follow uniform distribution within their corresponding uncertainty ranges, which were drawn from previous studies (e.g., Khorashadi Zadeh et al., 2017; Li et al., 2021; Mao et al., 2024), and/or recommended ranges in the Theoretical Documentation of SWAT (Neitsch et al., 2011). Sensitivity of these parameters were tested using a spatially lumped strategy, i.e., parameters were either assigned same values across the basin (such as snowfall temperature SFTMP) or adjusted by a multiplicative factor (such as all soil-related parameters) to maintain their spatial relationship (Table S2). Relative importance was assessed using the Morris method, with the Nash-Sutcliffe Efficiency (NSE) between observed and simulated monthly runoff at the ZJS gauge station for 1972 - 1986 (180-months) serving as the response metric:

$$\text{NSE} = 1 - \frac{\sum_{i=1}^{n} (Q_{o,i} - Q_{s,i})^2}{\sum_{i=1}^{n} (Q_{o,i} - \overline{Q_o})^2}, \tag{5}$$

where $n = 180$ is the number of months in the evaluation period; $Q_{s,i}$ and $Q_{o,i}$ are the SWAT simulated and observed runoff at month $i$, respectively; $\overline{Q_o}$ represent the mean observed runoff over the evaluation period.

A total of 500 trajectories were generated, resulting in $17,000 = 500 \times (33 + 1)$ SWAT simulations. For each run, NSE was computed between the simulated and observed 180-monthly runoff series. The Morris method was then used to calculate $\mu_i^*$ and $\sigma_i$ for each parameter, which reflect, respectively, the magnitude of a parameter's overall influence and the extent of interaction or nonlinearity. A scatterplot of $\mu_i^*$ versus $\sigma_i$ (Figure S2) was utilized to screen the influential parameters. This procedure screened five dominant parameters: CN2, CANMX, ESCO, CH_K1, RCHRG_DP. These parameters regulate key hydrological processes in SWAT, including surface runoff (CN2), canopy interception (CANMX), soil evaporation (ESCO), channel transmission losses (CH_K1), and groundwater recharge (RCHRG_DP). Descriptions of these parameters are





provided in Table 1. Following this screening step, these five parameters were retained for subsequent SSA and time-varying

SSA using Sobol' method as detailed in next section.

**Table 1: Description of the five parameters screened by the Morris method**

| Parameter | Input file extension | Description | Spatial parameterization | Range |
|---|---|---|---|---|
| CH_K1 | .sub | Effective hydraulic conductivity in tributary channel (mm/hr) | Subbasin | [0, 300] |
| CN2* | .mgt | Initial SCS runoff curve number for moisture condition II | Subbasin/HRU | [0.75, 1.25] |
| CANMX | .hru | Maximum canopy storage (mm $H_2O$) | Subbasin/HRU | [0, 100] |
| ESCO | .hru | Plant uptake compensation factor | Subbasin/HRU | [0, 1] |
| RCHRG_DP | .gw | Deep aquifer percolation fraction | Subbasin/HRU | [0, 1] |

*CN2 is adjusted by a multiplicative factor to account for spatial variability. The other four parameters are specified directly

per subbasin or HRU through the replacement method. All parameters are assumed to follow uniform distribution.

**2.5 Identification of Dominant Parameters via a Deep Learning-Assisted Sobol' method**

**2.5.1 Spatial Parameterization for Distributed Parameters**

In SWAT, parameters can be defined at three spatial levels: (1) basin-level parameters that remain a uniform value across the

entire basin, (2) subbasin-level parameters that vary among subbasins, and (3) HRU-level parameters that differ across

HRUs (Yu et al., 2018). This spatial parameterization allows parameter to be represented either as a coarse, spatially

aggregated value or as a detailed, spatially distributed one. From the Morris screening, five parameters were retained for

SSA and time-varying SSA (Section 2.4). Among them, parameter CH_K1 is inherently defined at the subbasin level,

allowing it to vary independently across the 39 subbasins. The other four parameters (CN2, CANMX, ESCO, and

RCHRG_DP) are more flexible in SWAT and can be parameterized either at the subbasin level or at the HRU level,

depending on the spatial resolution of analysis (Table 1).

To capture the influence of spatial scale on parameter sensitivity, the five parameters were defined under two distinct

parameterization schemes:

- Subbasin-scale parameterization. All five parameters were independently specified for each of the 39 subbasins,
  resulting in a total of 195 distributed parameters (5 parameters × 39 subbasins). This configuration reflects
  aggregated hydrologic responses at the subbasin level and provides a coarser yet computationally efficient

perspective of parameter influence.

- HRU-scale parameterization. CH_K1 remains specified per subbasin, while the other four parameters were
  independently defined across the 630 HRUs. This resulted in a total of 2,559 distributed parameters (1





parameters × 39 subbasins + 4 parameters × 630 HRUs). Although this substantially increased the number of parameters and, consequently, the dimensionality of the parameter space, it enabled a fine-grained characterization of heterogeneity in HRUs, i.e., land use, soil, and slope within subbasins.

Contrasting these two schemes clarifies how parameter influence aggregates at the subbasin level versus how it manifests locally at the HRU level. This design thus exposes scale effects on sensitivity "hotspots," informing both interpretation and scale-appropriate calibration strategies.

### 2.5.2 Spatial Parameterization for Distributed Parameters

Due to the computational impracticality of performing Sobol' method on such high-dimensional input spaces, particularly at the HRU scale, deep-learning surrogates were constructed to emulate SWAT efficiently (Razavi et al., 2012). The task is a high-dimensional, vector-to-vector regression: map a static parameter vector (length 195 or 2559) to a fixed-length sequence of 180-month runoff series at the ZJS gauge. Because both inputs and outputs are fixed size and temporal structure is encoded in the output vector, we selected fully connected MLPs rather than sequence models such as Long Short-Term Memory (LSTM) (Yang et al., 2024; Yu et al., 2024).

Two MLP surrogates were built to mirror the two spatial parameterizations:

- Subbasin-scale MLP surrogate. Comprised an input layer with 195 neurons (corresponding to 195 distributed parameters of subbasin-scale parameterization), four hidden layers with varying neurons, and an output layer with 180 neurons representing 180-month runoff values.

- HRU-scale MLP surrogate. Designed to accommodate higher spatial details, with 2,559 input neurons (corresponding to 2,559 distributed parameters of HRU-scale parameterization) and four hidden layers containing more neurons to capture the increased input dimensionality.

All hidden layers used Rectified Linear Unit (ReLU) activation functions, and dropout and batch normalization were selectively applied to mitigate overfitting and improve convergence stability. The overall architectures of both surrogate models are illustrated in Figure 2.

Training data were generated by sampling parameters from the ranges in Table 1 using Latin Hypercube Sampling (LHS) method. For the subbasin-scale surrogate, 50,000 parameter realizations were created, forming a 50,000 × 195 parameter matrix. For the HRU-scale surrogate, 200,000 realizations were generated, resulting in a 200,000 × 2,559 parameter matrix. These sample sizes balance (i) adequate coverage of the high-dimensional spaces needed for deep models and (ii) computational feasibility given the number of SWAT simulations. Preliminary experiments indicated only marginal accuracy gains beyond these sizes, confirming their adequacy. Each sampled parameter realization was fed to SWAT to obtain the paired 180-month runoff series, yielding input-output datasets for supervised learning.

All SWAT simulations were executed on the High-Performance Computing (HPC) platform of Northwest A&F University, which supports parallel batch job submission and thereby greatly reduced the total wall-clock time. The resulting input-output datasets were randomly divided into training (80%) and testing (20%) subsets. Both MLP surrogates were constructed



by using PyTorch (Paszke et al., 2019) and trained independently using the root-mean-squared error (RMSE) loss function, optimized using the Adam algorithm with initial learning rate of 0.001. To mitigate overfitting, early stopping was implemented using the validation loss as the stopping criterion. Performance was evaluated on the testing set using RMSE, the determination coefficient ($R^2$), and the predication error diagnostics. Once validated, the surrogates reproduced SWAT outputs with high fidelity while drastically reducing computational cost. These trained MLPs were subsequently used in place of SWAT for Sobol' method, enabling comprehensive, large-scale spatiotemporal assessments at both subbasin and HRU scales.

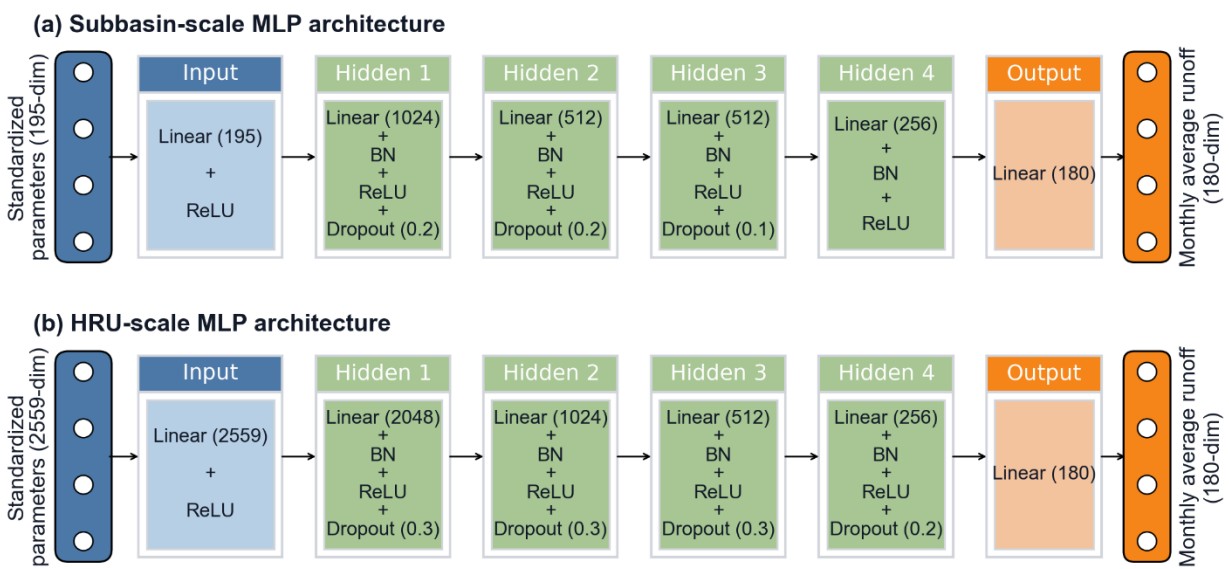

**Figure 2: MLP architectures for runoff prediction at the ZJS gauge station with spatial parameterization at (a) subbasin and (b) HRU scales. Numbers in parentheses after "Linear" give layer width (number of neurons); numbers after "Dropout" give the dropout rate. ReLU activation follows every Linear layer except the final output layer.**

### 2.5.3 Full-period SSA at Subbasin and HRU Scales

Using the trained surrogate models, SSA was conducted over the full 180-month simulation period to evaluate the spatial variability of parameter sensitivity. Unlike the Morris screening method implemented in Section 2.4, which relied on spatially lumped strategy, this SSA employed spatially disaggregated parameter definitions across 39 subbasins and 630 HRUs. Specifically, the five screened parameters were allowed to vary independently across all subbasins and HRUs, yielding 195 distributed parameters at the subbasin scale and 2,559 distributed parameters at the HRU scale, as detailed earlier. Then SSA using the Sobol' method was applied at both subbasin and HRU sales:

- Subbasin-scale SSA. Two sampling matrices **A** and **B**, each comprised 4,096 realizations, were first generated using Sobol' sequence. For each of the 195 parameters, an additional matrix $\mathbf{C}_i$ was constructed by replacing the $i$-th column of **A** with that of **B**. This resulted in a total of $806, 912 = 4,096 \times (195 + 2)$ parameter realizations.





- HRU-scale SSA. Due to the much higher input dimensionality, each base matrix **A** or **B** contained 32,768 realizations, and corresponding **C**$_i$ matrices were similarly generated. In total, this resulted in 83,918,848 = 32,768 × (2,559 + 2) parameter realizations.

For each parameter realization at either spatial scale, the corresponding MLP surrogate generated the 180-month runoff series at the ZJS gauge station. Consistent with practices in previous time-varying SSA studies (Herman et al., 2013b; Wu et al., 2022), sensitivity was assessed using summary performance metrics, i.e., NSE over the full 180-month simulation period, rather than raw streamflow values, since the objective was to evaluate the controls on overall model performance rather than on runoff alone.

The generation of sampling matrices and the calculation of Sobol' sensitivity indices were performed using the SALib Python package (Herman and Usher, 2017). First-order ($S_i$) and total-effect ($S_{Ti}$) sensitivity indices were computed for the 195 distributed parameters at the subbasin scale and the 2,559 distributed parameters at the HRU scale. In this study, the analysis primarily focused on the $S_{Ti}$ values because hey account not only for the direct effects of individual parameters but also for their interactions and nonlinear contributions. Given the strong nonlinearity and interaction structure of distributed hydrologic models such as SWAT, $S_{Ti}$ values are expected to provide a more comprehensive and reliable measure of parameter influence, thereby offering stronger guidance for model calibration and reduction.

For visualization, **subbasin-scale** $S_{Ti}$ values were reshaped into a 5 × 39 array (corresponding to five parameters defined across 39 subbasins) and mapped to the corresponding subbasins. This structure produced five geospatial maps, one for each parameter, with individual subbasins displaying the spatial distribution of parameter sensitivities. At the HRU scale, CH_K1 was mapped at the 39 subbasins, while the other four parameters (CN2, CANMX, ESCO, and RCHRG_DP) were mapped across the 630 HRUs. This yielded another five geospatial maps: one for CH_K1 at the subbasin scale and four for the remaining parameters at the HRU scale. Collectively, these spatial fields reveal the magnitude and location of influence and highlight sensitivity hotspots at each resolution.

### 2.5.4 Full-period SSA at Subbasin and HRU Scales

To capture temporal evolution in parameter influence, we performed time-varying SSA using the same parameter samples and surrogate outputs as Section 2.5.2; no additional surrogate evaluations were needed. Instead of computing indices based on the full-period NSE, rolling temporal windows were applied to form windowed NSE targets. This procedure yielded temporally resolved sensitivity indices for the distributed parameters, thereby constituting the time-varying SSA.

Two distinct temporal window lengths were selected to represent different hydrological variability scales:

- A 3-month window with 1-month step. The 3-month window emphasizes event to seasonal dynamics, allowing detection of rapid shifts in sensitivity linked to floods or droughts.
- A 24-month window with 6-month step. The 24-month window captures multi-year to interannual variations, emphasizing slower processes such as groundwater recharge, storage carryover, and long-term flow integration (Yilmaz et al., 2008; Zhang et al., 2011).





These settings produced 178 window intervals for the 3-month window and 27 window intervals for the 24-month window. In conjunction with the two spatial scales, this yielded four distinct spatiotemporal scales: (1) 3-month window at the subbasin scale, (2) 24-month window at the HRU scale, (3) 24-month window at the subbasin scale, and (4) 24-month window at the HRU scale. For each scale, Sobol' sensitivity indices were recomputed using the corresponding windowed

NSE. Take the 3-month window at the subbasin scale for example, the time-varying SSA analysis generated $178 \times 195$ $S_{Ti}$ values, corresponding to 178 window intervals and 195 distributed parameters. By remapping these $S_{Ti}$ values to the corresponding five screened parameters and subbasins, 178 time-ordered sensitivity maps were obtained for each parameter. These maps can be visualized as heatmaps (subbasins on the y-axis and window intervals on the x-axis) or assembled into animations that depict the spatiotemporal evolution of parameter influence (Herman et al., 2013a). The resulting temporally

resolved sensitivity fields delineate both spatial "hotspots" and temporal "hot moments" of sensitivity, revealing seasonal and interannual shifts in dominant hydrologic controls. Such dynamic insights provide a foundation for targeted parameterization, calibration, and monitoring strategies under changing hydroclimatic conditions.

## 3 Results

### 3.1 Performance of Deep Learning Surrogates

Figure 3 compares surrogate predictions against SWAT-simulated runoff at ZJS gauge station and illustrates the associated error structures via heatmaps. The performance of the subbasin- and HRU-scale MLP surrogates were evaluated on 10,000 and 40,000 independent test realizations, respectively; each comprising 180-month runoff series. Across these full test datasets, both surrogates accurately reproduced SWAT-simulated runoff dynamics, achieving global $R^2$ values of 0.9975 and 0.9981, and global RMSE values of 3.40 m³/s and 2.92 m³/s, respectively. These results indicate that the surrogates capture

the high-dimensional input-output mapping over the full 180-month horizon.

The error heatmaps further demonstrate the fidelity of the surrogates. For both surrogates, prediction errors (the difference between MLP-predicted and SWAT-simulated runoff) were generally symmetric and centered near zero, with 90.06% (subbasin-scale MLP surrogate; Figure 3a4) and 92.04% (HRU-scale MLP surrogate; Figure 3b4) of monthly errors falling within ±5 m³/s. The largest deviations were observed during high-flow months (Figures 3a2 and 3b2), reflecting the intrinsic

difficulty of emulating rapid hydrologic responses governed by threshold-like processes such as saturation-excess runoff, infiltration-excess generation, and in-channel routing. These deviations also stem partly from the limited representation of extreme events in the training dataset. Despite these challenges, the HRU-scale MLP surrogate achieved slightly lower RMSE and errors relative to the subbasin-scale MLP surrogate, consistent with its finer spatial resolution better capturing heterogeneity in infiltration capacity and storage dynamics that control stormflow generation.






**(a) Subbasin-scale MLP surrogate**

**(b) HRU-scale MLP surrogate**


**Figure 3:** **Performance of the trained MLP surrogates at the (a) subbasin and (b) HRU scales. Panels (a1) and (b1) compare ensemble SWAT simulations for the test cases (grey lines; 10,000 cases at the subbasin scale and 40,000 cases at the HRU scale), the specific SWAT-simulated runoff with lowest RMSE relative to observed runoff (black lines) and the corresponding MLP-predicted runoff (blue dashed lines), as well as observed runoff (black circles) at the ZJS gauge station. Panels (a2) and (b2) show heatmaps of prediction errors (MLP - SWAT) across 180 months from 1972 to 1986 for all test cases; blue solid lines indicate the RMSE values calculated for each month across realizations. Panels (a3) and (b3) present scatter plots of MLP-predicted versus SWAT-simulated runoff, showing strong agreement with global $R^2$ values of 0.9975 and 0.9981, and global RMSE values of 3.40 m$^3$ s$^{-1}$ and 2.92 m$^3$ s$^{-1}$, respectively. Panels (a4) and (b4) show the error density, which further indicate that 90.06% and 92.04% of**





**monthly prediction errors fell within ±5 m³ s⁻¹ for the subbasin- and HRU-scale MLP surrogates, respectively, confirming high predictive fidelity.**

From a computational standpoint, the efficiency gains through the MLP surrogates were transformative. A single SWAT simulation takes ~10 seconds of CPU time on a 2.2 GHz Intel® Core™ i7-14650HX laptop. In contrast, the trained subbasin-scale MLP surrogate completed 806,912 evaluations in ~2 seconds, corresponding to ~2.5×10⁻⁶ s per evaluation,

and the HRU-scale MLP surrogate complete 83,918,848 evaluations in ~475 seconds, corresponding to ~5.7×10⁻⁶ s per evaluation. These correspond to speed-ups ~4.0 × 10⁶ and ~1.8 × 10⁶, respectively, i.e., efficiency gains exceeding six orders of magnitude relative to the original SWAT model. Such acceleration renders SA at multiple spatial and temporal scales computationally tractable. Additionally, this remarkable efficiency enables rapid parameter screening and supports downstream applications such as real-time forecasting, model-data fusion, and optimization-based decision support within

distributed hydrologic modelling frameworks.

### 3.2 Spatial Distribution of Parameter Sensitivities at Subbasin and HRU Scales

Figure 4 maps the spatial distribution of Sobol's total-effect sensitivity index $S_{Ti}$ for five screened parameters using the full-period NSE as the sensitivity response by both subbasin-scale and HRU-scale SSA (results of the first-order sensitivity index $S_i$ exhibit similar spatial patterns and are provided in Figure S3). The maps highlight the spatial heterogeneity of parameter

influence across the basin, while the bar plots beneath each map rank the ten most sensitive subbasins or HRUs. Note that CH_K1 can only be defined at the subbasin level (Table 1) and was thus analysed solely at that scale, even when included in HRU-scale SA.

At the subbasin scale (Figure 4a), parameter sensitivity was unevenly distributed across the subbasins, and disproportionately high values concentrated near the ZJS gauge station (outlet of subbasin 36; black square). Subbasins 33

and 34 consistently emerged as the sensitivity hotspots across all five parameters, underscoring their dominant role in controlling runoff variability. This pattern seems underscore the leverage of gauge-proximal control volumes on the downstream hydrograph, whereby modest parameter perturbations in these locations propagate efficiently to the gauged signal, which has also been reported by Tang et al. (2007). Several additional subbasins, including 14, 21, and 30, and occasionally upstream subbasins 2 and 7, also ranked among the top ten for specific parameters, reflecting localized

dominance of particular hydrologic processes. In contrast, the three terminal subbasins (37 - 39) located downstream of the gauge exhibited $S_{Ti}$ values near zero, indicating negligible influence on upstream-controlled discharge dynamics.





**(a) Spatial distribution of Sobol's $S_{Ti}$ by subbasin-scale SSA**

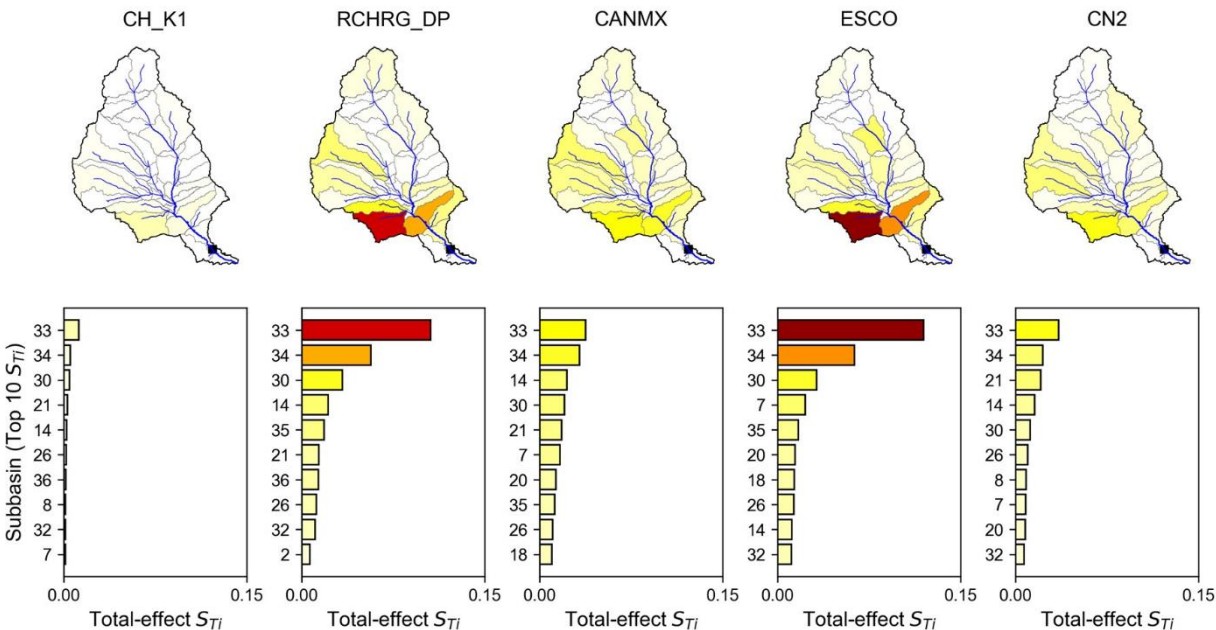

**(b) Spatial distribution of Sobol' $S_{Ti}$ by HRU-scale SSA**

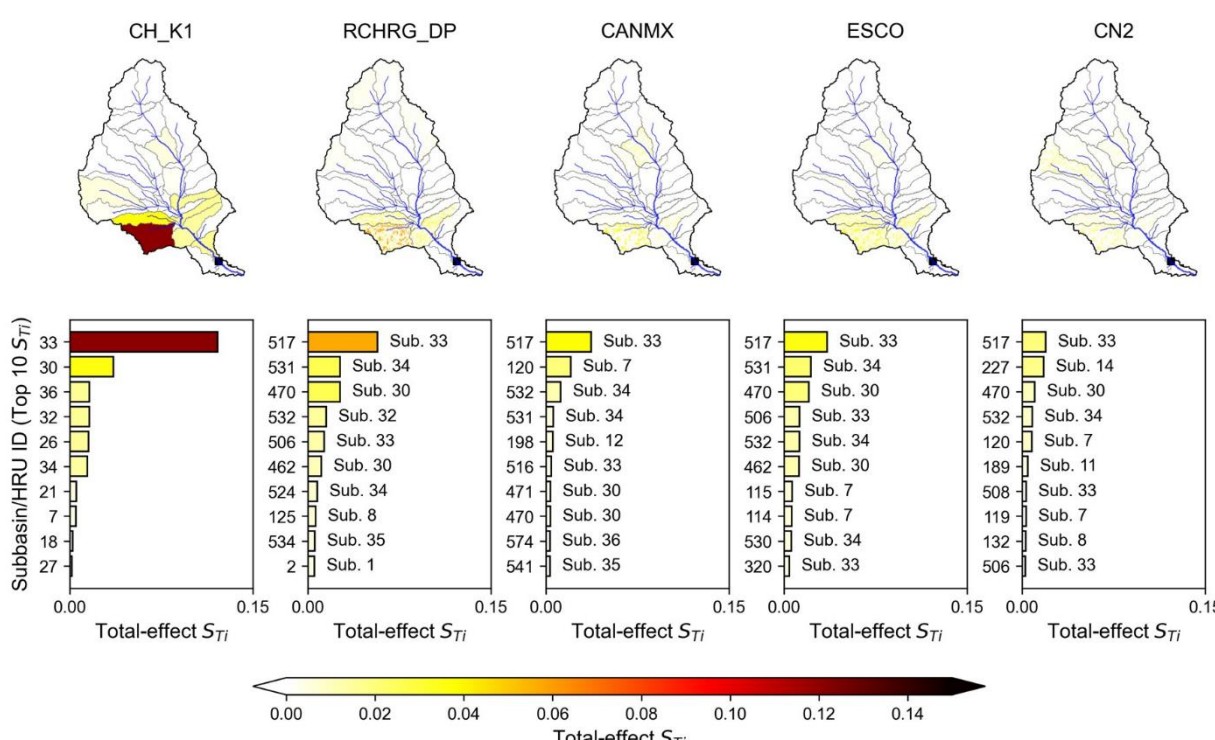





**Figure 4: Spatial distribution of Sobol's total-effect $S_{Ti}$ sensitivity index based on the full-period NSE by (a) subbasin-scale SSA and (b) HRU-scale SSA. Bar plots below each map show the top ten subbasins/HRUs ranked by $S_{Ti}$. For the HRU-scale plots, the text to the right of each bar indicates the parent subbasin (e.g., HRU 517 belongs to subbasin 33, denoted as Sub. 33).**

At the HRU scale (Figure 4b), the $S_{Ti}$ values remain highly uneven across both subbasins and HRUs. Most regions, particularly those in the upper basin, exhibit near-zero $S_{Ti}$ values, indicating that the negligible influence on the gauge's hydrograph. This may partly arise from the substantial increase in parameter dimensionality at the HRU level, which disperses the total variance contribution among a much larger number of parameters. Additionally, three scale-dependent features stand out. First, the apparent influence of CH_K1 increases. This behavior reflects a scale-interaction mechanism: as the other four parameters (CN2, ESCO, CANMX, and RCHRG_DP) are disaggregated to individual HRUs, the routing control exerted by CH_K1 operates on the aggregate of numerous fast, spatially heterogeneous responses, thereby elevating its relative contribution to the overall model performance. A complementary explanation aligns with Acero Triana et al. (2025), who reported that the sensitivities of channel routing parameters are particularly affected by spatial discretization and tend to become more relevant as model resolution becomes finer.

Second, sensitivity patterns become more localized within subbasins. Although the most sensitive HRUs remain concentrated in subbasins 33 and 34, their responses display pronounced heterogeneity. For example, HRU 517 in subbasin 33 (HRU IDs 505 - 521; see Table S1) ranks among the most sensitive units for multiple parameters, whereas others within the same subbasin show minimal sensitivity. Such intra-subbasin contrasts underscore the influence of soil, topographic, and land-cover heterogeneity, which becomes blurred when parameters are aggregated at the subbasin scale.

Third, a phenomenon of partial scale decoupling is observed. Some HRUs exhibit high sensitivity for specific parameters even when their parent subbasins do not appear among the top ten at the subbasin scale, e.g., HRU 198 in subbasin 12 for CANMX. This decoupling indicates that spatial aggregation can obscure localized controls, while HRU-scale diagnostics provide additional, actionable insights for calibration and monitoring. Collectively, these findings support a hierarchical calibration strategy: prioritizing gauge-proximal subbasins to constrain basin-scale responses, and within them, targeting sensitive HRUs to refine parameter ranges associated with dominant local processes.

### 3.3 Temporal Variability of Parameter Sensitivities at Multiple Spatiotemporal Scales

Animations in the Supporting Information display the temporal evolution of spatially distributed $S_{Ti}$ for the five screened parameters at both spatial resolutions (Movie S1 for 3-month window and Movie S2 for 24-month window, https://figshare.com/articles/media/Movies_S1_and_S2/30634316). For interpretability, Figure 5 summarizes subbasin-scale results as heatmaps for the 3-month window (Figure 5a) and the 24-month window (Figure 5b). In each panel, the left heatmap shows the spatiotemporal distribution of $S_{Ti}$, with the y-axis representing the 39 subbasins and the x-axis showing the corresponding window intervals (178 intervals for the 3-month window and 27 for the 24-month window). Overlaid black lines are the corresponding normalized moving average runoff values at the ZJS gauge station, scaled within 0 - 1 using min-max normalization with the original range of 16.29 m³/s to 179.74 m³/s for 3-month window and 40.10 m³/s to





75.85 m$^3$/s for 24-month window. These moving average runoff lines provide important hydrological context. The right heatmaps show lagged Spearman's rank correlations ($r$) between $S_{Ti}$ and runoff over all intervals for each subbasin within [-2, +2] lag intervals. Additionally, statistically significant correlations ($p < 0.05$) are marked with crosses, while blank cells denote subbasins where $S_{Ti}$ was consistently below 0.001 across all intervals. The use of lagged correlations is to capture the

potential time shifts between the occurrence of sensitivity peaks in model parameters and the corresponding changes in runoff. Negative lag values suggest that runoff precedes parameter sensitivity and positive lag values indicate that parameter sensitivity precedes runoff. When considered alongside the Spearman's rank correlation coefficient $r$, this analysis enables the detection of dynamic relationships between runoff and parameter sensitivity. For example, a significant positive correlation with a negative lag of one interval suggests that an increase in runoff may lead to a corresponding increase in

parameter sensitivity over time, but with a delayed response of parameter sensitivity by one interval, possibly due to runoff-driven changes in the environment or model behavior. Two structural features persist across both window lengths: (1) subbasins 33 and 34 consistently emerge as sensitivity hotspots across all parameters, and (2) subbasins 37 - 39, located downstream of the gauge, maintain near-zero $S_{Ti}$, confirming they do not affect the upstream hydrograph.

With the 3-month window at subbasin scale (Figure 5a), $S_{Ti}$ values fluctuate sharply with hydrologic events, showing

intermittent bursts synchronized with high- or low-flow phases, emphasizing the transient nature of fast hydrological processes. CN2 displays distinct peaks of $S_{Ti}$ during high flows, typically with positive correlations near zero lag, indicating a contemporaneous link between storm runoff generation and curve-number controls. In contrast, RCHRG_DP, which controls the proportion of water percolating to the deep aquifer, demonstrates an inverse relationship with runoff, as indicated by negative correlations. This is consistent with higher deep-percolation fractions depleting shallow storage and

reducing baseflow, especially in dry periods. CANMX, the maximum canopy storage, shows episodic increases during wet spells, with sensitivity peaks lagging slightly (by one or two intervals) behind high-flow events. This lag indicates that CANMX becomes important when the canopy approaches saturation. Once rainfall fills the canopy storage, small variations in CANMX can substantially influence the amount of throughfall reaching the ground, resulting in higher sensitivity following high-flow events. ESCO, the soil evaporation compensation factor, shows a weaker but similar pattern, with

sensitivity increasing during wetter conditions, indicating its role in controlling evapotranspiration as soils become wet. CH_K1, the channel permeability, shows sporadic sensitivity that is not as persistent in time, reflecting its limited influence at event scale.

When the window length is extended to 24 months (Figure 5b), the sensitivity patterns become more persistent, reflecting the importance of longer-term storage and routing processes. The spatial hotspots identified at the 3-month windows (e.g.,

subbasins 33 and 34) remain highly sensitive at the 24-month windows, suggesting that these locations exert sustained control on basin response through cumulative flow accumulation and storage effects. However, for CN2, a distinct contrast emerges between upstream and downstream regions. In upstream subbasins such as 2, 5, and 6, and midstream subbasins such as 20 and 21, periods of elevated CN2 sensitivity precede increases in runoff, resulting in negative correlations at positive lags. In contrast, downstream subbasins like 33 and 34 generally show positive correlations, indicating that higher



$S_{Ti}$ values are associated with increased surface runoff. This spatial shift underscores the role of hydrological connectivity and routing, where upstream infiltration–storage dynamics gradually integrate into downstream runoff responses through flow accumulation and storage buffering across the catchment. Such contrasts likely reflect differences in hydrological connectivity, with transitional midstream subbasins (e.g., 20 and 21) responding earlier through infiltration–storage processes, while other midstream areas with weaker connectivity or greater storage capacity show less synchronized

responses. For RCHRG_DP, sensitivity at the 24-month window becomes more pronounced, particularly in subbasins 33 and 34, where $S_{Ti}$ values exhibit strong positive correlations with runoff. This pattern contrasts with the 3-month window, where RCHRG_DP is more sensitive under low-flow conditions. The contrast between temporal scales indicates that RCHRG_DP exerts both short-term control on recharge losses and long-term influence on groundwater storage and delayed flow contributions. At the interannual scale, sustained deep percolation enhances aquifer storage and subsequently supports high-

flow recovery, whereas at shorter timescales it primarily regulates immediate recharge–runoff partitioning. ESCO also exhibits a broader and more persistent temporal influence, with predominantly positive correlations concentrated in subbasins 33 and 34. This pattern reflects its role in modulating evapotranspiration over both wet and dry periods, thereby integrating short-term variability into longer-term water balance. In contrast, CH_K1 and CANMX show weaker and more spatially fragmented correlations at the 24-month window, suggesting that routing transients and canopy storage thresholds

which are important at event or seasonal timescales, become less influential once runoff responses are aggregated over longer hydrologic periods.







**Figure 5: Time-varying Sobol's total-effect indices $S_{Ti}$ for the five parameters at the subbasin scale using (a) a 3-month moving window with 1-month timestep and (b) a 24-month moving window with 6-month timestep. The left heatmaps of each figure show the spatiotemporal distribution of $S_{Ti}$, with subbasins ordered by ID on the y-axis and time on the x-axis. The black line overlays indicate the corresponding normalized moving-mean runoff at the ZJS gauge station. The right heatmaps show lagged Spearman's rank correlations ($r$) between and runoff, where the x-axis represents lag in intervals (negative = runoff leads; positive = sensitivity leads). Crosses (×) denote statistically significant correlations at $p < 0.05$. For subbasins with $S_{Ti}$ remained below 0.01 across all window intervals, correlations were not calculated and are shown as blank cells.**

Figure 6 presents the temporal evolution of $S_{Ti}$ for the five parameters at the HRU scale using the 3-month (Figure 6a) and 24-month (Figure 6b) windows with corresponding steps. As at the subbasin scale, the 3-month windows reveal burst-like sensitivity aligned with high- and low-flow phases, whereas the 24-month windows suppress short-term variability and highlight persistent, longer-term controls. Across both window lengths, CH_K1 is visibly more dominant in space and time than the other parameters. This is expected, as discussed in Section 3.2, because CH_K1 is defined at the subbasin scale and operates on the aggregate of many fast, heterogeneous HRU responses, thereby elevating its relative contribution. The





generally smaller $S_{Ti}$ magnitudes for CN2, CANMX, ESCO, and RCHRG_DP reflect variance dilution across the much larger HRU-scale parameter set.

As shown in Figure 6, the hotspots and hot moments of $S_{Ti}$ are generally align with those observed at the subbasin scale (Figure 5). However, the HRU-scale analysis provides a more granular view of parameter sensitivity, revealing spatial heterogeneity that is often masked in aggregated subbasin results. For example, while subbasins 33 and 34 consistently emerged as sensitivity hotspots at the subbasin scale (Figure 5), they do not exhibit uniformly high $S_{Ti}$ across all their HRUs. Instead, only a subset of HRUs (e.g., HRU 517 in subbasin 33) within these subbasins show pronounced sensitivity, while

neighboring HRUs remain near zero. This suggests that the aggregated sensitivity at the subbasin scale is disproportionately driven by a limited number of responsive HRUs, rather than by uniform contributions across the entire subbasin.

Additionally, the lagged Spearman's correlations at the HRU scale align with those observed at the subbasin scale (Figure 5). However, at the HRU scale, these correlations reveal additional spatial variability. Notably, there are differences in the timing and strength of parameter–runoff interactions between individual HRUs within the same subbasin. This highlights the

importance of HRU-scale analysis, which provides a more refined understanding of parameter influence. It is crucial for linking hydrological processes to land surface characteristics and management practices, and it supports more targeted model calibration and monitoring strategies.





**Figure 6: Time-varying Sobol' total-effect indices $S_{Ti}$ for the five parameters at the HRU-scale analysis using (a) a 3-month moving window with 1-month timestep and (b) a 24-month moving window with 6-month timestep. The left heatmaps show the spatiotemporal distribution of $S_{Ti}$, with HRUs ordered by ID on the y-axis and time on the x-axis. The black line overlays indicate the corresponding moving-mean runoff at the ZJS gauge station. The right heatmaps show lagged Spearman's rank correlations ($r$) between and runoff, where the x-axis represents lag in intervals (negative = runoff leads; positive = sensitivity leads). Crosses**
**(×) denote statistically significant correlations at $p < 0.05$. For HRUs with $S_{Ti}$ remained below 0.001 across all window intervals, correlations were not calculated and are shown as blank cells.**

## 4 Discussion

### 4.1 Effects of Scales on Identification of Sensitivity Hotspots and Hot Moments

Figure 4 illustrated the parameter sensitivities exhibit pronounced spatial heterogeneity across subbasins and HRUs. To
elucidate the effects of scales on this heterogeneity, we computed the sum of Sobol's total-order sensitivity indices $S_{Ti}$ of the




five screened parameters, $\Sigma S_{Ti}$, within each spatial unit (Figures 7a and 7b). Specifically, the five spatial maps of $S_{Ti}$ obtained from subbasin- and HRU-scale SSA (Figure 4) were overlaid and summed to derive the spatial distribution of $\Sigma S_{Ti}$ at both scales. Because CH_K1 is only defined at the subbasin scale, the CH_K1 is equally distributed to the HRUs in the subbasins when estimate the $\Sigma S_{Ti}$ at the HRU scale. Because $S_{Ti}$ quantifies the total contribution of parameter $i$ to output variance, $\Sigma S_{Ti}$

serves as a compact measure of a unit's overall responsiveness to the model response, that is, its potential to transmit parameter perturbations into variations of the full-period NSE at the ZJS gauge. Larger $\Sigma S_{Ti}$ values are therefore more dynamically linked to the modeled runoff response and represent spatial sensitivity hotspots.

Figures 7a and 7b shows that the broad spatial organization of sensitivity is consistent across resolutions, while HRU-level results sharpen localization. For example, subbasins 33 and 34 (and to a lesser extent 30) consistently emerge as sensitivity

hotspots at the subbasin scale (Figure 9a). At the HRU scale (Figure 9b), the dominant HRUs are nested within these subbasins, e.g., HRU 517 (in subbasin 33), HRUs 531 and 532 (in subbasin 34), and HRU 470 (in subbasin 30). Thus, increasing spatial resolution refines where sensitivity concentrates rather than changing which regions matter most. This refinement reflects heterogeneity in land cover, soils, and slope within subbasins: only a subset of HRUs, often specific land-use/soil/slope combinations, drives most of the total sensitivity.

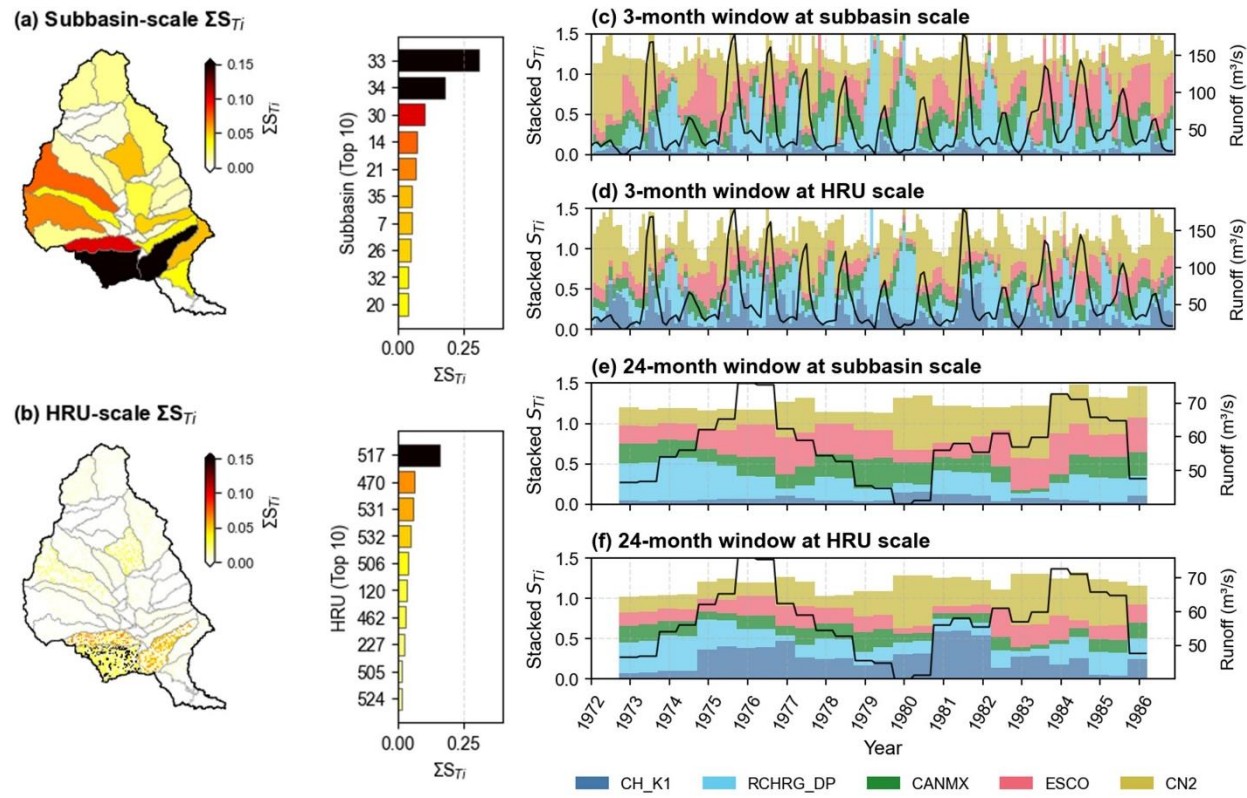


**Figure 7**: Spatial distribution of (a) the sum of Sobol' total indices, $\Sigma S_{Ti}$, of the five screened parameters at the subbasin scale; (b) $\Sigma S_{Ti}$ at the HRU scale; temporal variability of the summed $S_{Ti}$ for (c) the 3-month windows at the subbasin scale, (d) the 3-month windows at the HRU scale, (e) the 24-month windows at the subbasin scale, and (f) the 24-month windows at the HRU scale,





To explore how these sensitivity hot monuments shifts under different spatiotemporal scales, the $S_{Ti}$ of five screened
parameters was calculated using both 3- and 24-month moving windows at both spatial scales, as shown in Figures 7c – 7f.
For each window, $S_{Ti}$ was summed over 39 subbasins for CH_K1 and over 630 HRUs for the remaining four parameters;
stacked areas display the normalized composition of total sensitivity, with moving-average runoff at ZJS overlaid (black
line). At the 3-month windows (Figures 7c and 7d), parameter sensitivities oscillate seasonally in phase with runoff. CN2
peaks during wet months, reflecting infiltration-excess runoff control, with episodic contributions from CANMX via canopy
interception. ESCO and RCHRG_DP strengthen during dry phases, consistent with evapotranspiration partitioning and
recharge. The timing of these hot moments is nearly identical between subbasin and HRU aggregations, spatial resolution
primarily affects magnitude and granularity, not phase. At the 24-month windows (Figures 7e and 7f), event-scale
oscillations dampen and sensitivities evolve toward a smoother, persistent composition. CN2 remains dominant after
temporal averaging, while ESCO and RCHRG_DP provide stable secondary influence. CH_K1 stays modest at the subbasin
scale but becomes more evident when aggregated alongside HRU-resolved parameters, reflecting channel-hillslope linkages
expressed over longer timescales.

Overall, these findings suggest spatial scaling sharpens localization of hotspots without altering the catchment-scale ordering
of influential regions, while temporal scaling reshapes the relative weights and persistence of parameter effects. Across all
space-time combinations, CN2 is consistently most influential, with ESCO and RCHRG_DP providing robust secondary
controls and CH_K1 expressing localized effects tied to routing. This cross-scale stability is beneficial for calibration and
interpretation: parameter importance inferred at one resolution is largely transferable to the other, while finer spatial
discretization improves where, rather than whether, controls manifest.

**4.2 Implications, Limitations, and Future Study**

This study highlights that parameter sensitivity in distributed hydrologic models is scale-dependent, suggesting a need to
rethink traditional lumped calibration strategies. The findings imply that calibration should explicitly consider the spatial
distribution of parameters and their variable influence across hydrologic scales. Specifically, the results support a
hierarchical, cross-scale calibration approach in which parameter adjustment proceeds from localized HRU sensitivities to
aggregated subbasin behavior and finally to basin-scale evaluation. Such an approach can better constrain fine-scale
processes, reduce equifinality, and maintain consistency between distributed parameters and emergent hydrologic responses.
Moreover, maintaining parameters as spatially distributed fields (rather than aggregating them into lumped values) enables
the model to preserve heterogeneity in soils, land use, and topography while reflecting physically meaningful spatial
patterns. When combined with sensitivity-based hotspot and hot-moment diagnostics, this framework can guide targeted
calibration and monitoring, directing computational and observational efforts to locations and periods of highest model
responsiveness.



Despite its advantages, several limitations should be acknowledged. First, the parameter ranges in this study were derived from prior literature rather than posterior distributions constrained by calibration data. Although this approach ensures broad coverage of the feasible parameter space, it may not fully capture sensitivities under calibrated or equifinal conditions (Wu et

al., 2022). Second, model performance was assessed primarily using the NSE, which emphasizes peak flows but underrepresents variability and low-flow conditions. Employing additional criteria such as Kling-Gupta Efficiency (KGE), percent bias (PBIAS), or likelihood-based functions would yield a more comprehensive assessment of model behavior (Acero Triana et al., 2025; Tuo et al., 2018). Finally, while the surrogate models substantially reduce computational cost, they may introduce approximation bias in sparsely sampled regions of parameter space, especially for nonlinear interactions.

Future research should focus on translating the spatial and temporal sensitivity information into scale-aware calibration strategies. Integrating posterior parameter distributions obtained through Bayesian inference or Markov Chain Monte Carlo (MCMC) methods would ensure that sensitivity estimates reflect realistic parameter uncertainty (Zheng and Han, 2015). Developing multi-objective calibration frameworks that balance multiple performance metrics could better represent trade-offs between high- and low-flow dynamics. Additionally, incorporating data assimilation and active learning techniques can

refine surrogate accuracy and adaptively target high-sensitivity regions of parameter space. Ultimately, advancing toward spatiotemporally informed, distributed calibration frameworks will enhance model realism, reduce equifinality, and transform sensitivity analysis from a diagnostic exercise into a practical guide for robust, process-based parameter estimation.

## 5 Conclusion

This study presents an efficient framework to quantify how parameter sensitivity in distributed hydrologic models varies through space and time. Using SWAT for the Jinghe River Basin, we first applied a Morris screening with a spatially lumped strategy to identify influential parameters. We then trained two separate multilayer perceptron surrogates that preserve SWAT's spatial parameterization at both the subbasin and HRU levels, and computed time-varying Sobol' total-effect indices under short (3-month) and long (24-month) moving windows. The framework makes fine-grained, spatiotemporal

sensitivity analysis tractable while retaining diagnostic transparency.

Methodologically, the dual-scale surrogate approach is central: by emulating SWAT at the different spatial scales (subbasin vs. HRU) where parameters are defined, it allows a controlled assessment of how spatial discretization alters sensitivity structure. Coupling these surrogates with time-varying SSA extends static "maps" into "movies" revealing not only where sensitivity resides (hotspots) but also when it intensifies (hot moments) as temporal aggregation changes from event to

interannual scales. Practically, these movies translate directly into scale-aware calibration guidance: they indicate which parameters to prioritize at a given window length and where to focus effort within the basin, while acknowledging HRU-level heterogeneity inside sensitive subbasins.





Diagnostically, the results show that parameter sensitivity is strongly organized in both space and time. Spatially, hotspots cluster in hydrologically active, outlet-proximal subbasins (notably 33 and 34), consistent with large variability in surface

runoff, evapotranspiration, and channel transmission losses. At the HRU scale, those same hotspots persist but resolve into a small number of dominant HRUs embedded within sensitive subbasins, confirming that aggregation masks intra-subbasin heterogeneity. Temporally, the 3-month window exhibits pronounced seasonal oscillations synchronized with precipitation, with CN2 (and episodically CANMX) governing event-scale dynamics, while ESCO and RCHRG_DP strengthen during drydowns. When aggregated to 24 months, oscillations damp and the sensitivity composition stabilizes: CN2 remains

influential but less variable, whereas ESCO and RCHRG_DP express more persistent, storage-related control. CH_K1 is secondary at the subbasin scale yet becomes more apparent at the HRU scale, reflecting heightened channel-hillslope interactions once spatial heterogeneity is made explicit. Together, these patterns indicate that temporal aggregation reshapes parameter importance more than it changes hotspot locations: where sensitivity resides is stable; which parameter dominate is window-dependent.

These insights carry direct implications for practice. Calibration should be staged by timescale and location: use short-window diagnostics to prioritize CN2 (and secondarily CANMX) for matching peak timing and magnitude; use long-window diagnostics to refine ESCO and RCHRG_DP for interannual balance, baseflow, and recession behavior; and evaluate CH_K1 especially at HRU resolution where channel losses interact with heterogeneous hillslopes. Parsimony should be scale-aware, focusing parameters where and when they matter (outlet-proximal hotspots; HRU-level dominant

units) rather than applying uniform simplifications that ignore spatial and temporal structure.

In sum, the proposed two-step deep leanring-assisted time-varying SSA (i) provides a computationally feasible path to interrogate distributed models across spatiotemporal scales; (ii) explains why sensitivity hotspots persist in space while process dominance shifts with temporal aggregation; and (iii) offers actionable guidance for scale-consistent calibration. The approach is transferable beyond SWAT and, by clarifying how distributed parameters jointly control runoff through space

and time, can improve both diagnosis and prediction under nonstationary hydroclimatic conditions.

**Code and Data Availability**

The code and data used for this study will be made available on GitHub via https://github.com/jyangfsu/JRB_Spatiotemporal_Sensitivity and Zendo via https://doi.org/10.5281/zenodo.17627546 upon publication of the paper.

**Author Contributions**

J. Yang led the conceptualization, methodology design, model development, data analysis, visualization, and manuscript writing. J. Zhang contributed to the development of the sensitivity analysis framework and provided guidance on parameter




ranges used in the sensitivity analysis. T. Jiao supervised the study, contributed to the research design, and supported manuscript revision. Y. Zhao co-supervised the project, contributed to methodology refinement, and guided result interpretation. M. Luo assisted with data processing, model implementation, and figure preparation. L. Wu contributed to hydrological model setup, field data interpretation, and manuscript editing. M. Ye provided methodological guidance, contributed to the deep learning surrogate modelling strategy, and supported manuscript revision. J. Song contributed to critical manuscript review. All authors discussed the results and approved the final version of the manuscript.

## Competing Interests

The authors declare that they have no conflict of interest.

## Acknowledgements

This study was supported by the National Natural Science Foundation of China (Grant Nos. 42302288, 42402245, and 42277073), China Postdoctoral Science Foundation (Grant Nos. 2025T180086 and 2024M752619), Innovation Capability Support Program of Shaanxi (Grant No. 2024RS-CXTD-55), and Xi'an Young Talent Support Program (Grant No. 0959202513082). We thank the High-Performance Computing Platform of Northwest A&F University for providing computational resources.

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
