# Peer review of "Identifying Dominant Parameters Across Space and Time at Multiple Scales in a Distributed Model Using a Two-Step Deep Learning-Assisted Time-Varying Spatial Sensitivity Analysis"

_EGUsphere, 2025_

## Author Comment (AC2)

**Reply to RC2's Comment on egusphere-2025-5694**

**General Assessment**

The paper tackles an important problem, but several method choices and interpretations need improvement, and some conclusions appear over-claimed. I would recommend major revision.

**Reply:** We thank the reviewer for this assessment and the constructive comments, as well as for the opportunity to revise the manuscript. In the revised manuscript, we will address these issues by refining the methodological descriptions, strengthening robustness analyses where appropriate, and tempering interpretations to better reflect the underlying assumptions and limitations.

**Major Comments**

**Comment 1:** The authors state that no model calibration was performed to maintain "diagnostic integrity". While this isolates parameter sensitivity from calibration bias, it risks performing SA on a model that does not represent the physical reality of the Jinghe River Basin. Will sensitivity patterns change significantly once the model is constrained into a realistic posterior parameter space? You might want to report baseline SWAT performance vs observations and discuss how poor/mediocre fit would distort SA.

**Reply:** We thank the reviewer for raising this important point. In the original submission, sensitivity analysis was conducted using prior parameter ranges derived from the SWAT documentation and previous studies, with the intention of diagnosing process controls under a broadly defined parameter space rather than within a calibrated posterior. However, we acknowledge that the absence of calibration limits the extent to which the identified sensitivities can be interpreted as basin-specific physical controls.

Following the reviewer's suggestion, we first ran the SWAT model using the default parameter set. The resulting simulation shows very poor agreement with the observed runoff, with a full-period NSE of -6.0. We then examined model performance across the 17,000 simulations generated during the Morris screening stage. As illustrated in Figure R1, we compare the default SWAT simulation (baseline), the full ensemble of Morris simulations, the best-performing simulation within the ensemble (NSE = 0.70), and the observed runoff. This comparison highlights that, although the default parameterization performs poorly, the prior parameter space explored during the screening stage contains parameter combinations capable of reproducing the observed runoff reasonably well.

[Figure]

**Figure R1.** Comparison of the default SWAT simulation, the ensemble of 17,000 Morris simulations, and the observed runoff.

In the revised manuscript, we will therefore take two additional steps to address this issue. First, we will report baseline model performance against observations to provide appropriate context for the sensitivity results. Second, we will explicitly frame the sensitivity analysis as conditional on the assumed prior parameter space and further examine the effect of constraining this space using performance-based criteria. Specifically, we will restrict the parameter samples to those yielding, for example, a full-period NSE greater than 0.5 and reassess the resulting sensitivity patterns, including the identified hot spots and hot moments.

This additional analysis will allow us to evaluate how sensitivity magnitudes and spatiotemporal patterns change when the parameter space is conditioned on acceptable model performance. The interpretation of the sensitivity results will be adjusted accordingly to reflect their dependence on the assumed parameter space and to avoid over-claiming basin-specific physical realism.

**Comment 2:** The author chose fully connected MLPs over sequence models like LSTMs arguing that the temporal structure is "encoded in the output vector". However, hydrologic processes are inherently autoregressive. Would not a 180-neuron output layer treat each month runoff as an independent regression target and potentially ignoring the temporal dependencies and mass balance continuity inherent in the SWAT model?

**Reply:** We fully agree that hydrologic processes are inherently autoregressive and that sequence models such as LSTMs are, in principle, well suited to represent temporal dependencies. Recurrent architectures were therefore also considered during the early stages of this work.

The choice of a fully connected MLP in this study is driven by the specific problem formulation. All model parameters are time-invariant, and the meteorological forcing is identical across parameter samples. Under this setting, the surrogate approximates a deterministic mapping from a static parameter vector to the complete runoff time series simulated by SWAT. The 180-dimensional output thus represents a single structured response, rather than independent monthly regression targets. Although the MLP does not explicitly model temporal recurrence, the temporal

dependencies inherent in the SWAT simulations are implicitly captured through joint learning of the output vector.

We acknowledge that this rationale was not sufficiently clear in the original manuscript. In the revised version, we will clarify the assumptions under which the MLP formulation is appropriate and explicitly note that sequence models would be more suitable when time-varying parameters, dynamic forcing, or explicit state evolution are considered.

**Comment 3:** You find hotspot subbasins near the gauge (e.g., 33–34) dominate sensitivities. This is unsurprising when the response is a single-gauge performance metric and proximity/connectivity to the gauge will mechanically increase leverage as briefly acknowledged by the authors. However, the authors interpret patterns as reflecting spatial heterogeneity (land use/soil/topography). Would hotspot locations remain under alternative responses and/or multi-site constraints? You might want to discuss this including the effect of additional gauges, internal variables (ET, soil water, baseflow index), or alternative spatially distributed responses.

**Reply:** We thank the reviewer for highlighting this important point and fully agree that the choice of a single-gauge performance metric can introduce a spatial leverage effect, whereby subbasins closer to or more strongly connected with the outlet gauge exert a disproportionate influence on the sensitivity patterns. While this effect is briefly acknowledged in the manuscript, we agree that its implications should be discussed more explicitly.

We expect that hotspot locations would change under alternative response definitions or multi-site constraints. However, no additional streamflow gauges are available in the present case study to directly test this effect. To address this limitation, we will extend the Discussion to consider how sensitivity hotspots may differ when alternative, spatially distributed response variables are used. In particular, we will discuss the potential role of internal model variables (e.g., ET, soil water storage, and baseflow-related metrics) that are less directly tied to outlet proximity.

**Comment 4:** For the HRU-scale Sobol' analysis the authors used k = 2559 and N = 32768. The authors noted that sensitivity magnitudes ($S_T$) are smaller at this scale due to "variance dilution" Is the identified sensitivity a physical signal or a mathematical artifact of the Sobol' method when the input space is massively expanded? With thousands of parameters, many $S_T$ estimates can be noisy/biased; negative indices and non-closure can occur and should be diagnosed. You might want to include convergence checks including reporting fraction of negative indices and demonstrating stability of rankings/hotspots. Otherwise, soften your claims and frame findings as exploratory.

**Reply:** We thank the reviewer for this important and technically insightful comment. We fully agree that Sobol' sensitivity analysis in very high-dimensional parameter spaces raises concerns regarding estimator noise and potential numerical artifacts, including small or negative sensitivity estimates and imperfect variance closure.

Following the reviewer's suggestion, we will explicitly diagnose these issues in the revised manuscript. We will include convergence checks and additional diagnostics in the Supporting Information, including (i) the stability of $S_T$ values with increasing sample size, (ii) the fraction and magnitude of negative sensitivity estimates, and (iii) indicators of variance closure. These

analyses will allow us to assess whether the identified HRU-scale sensitivities are numerically robust and to what extent they can be interpreted as physically meaningful.

Based on these diagnostics, we will refine the interpretation of the HRU-scale results, clearly distinguishing robust, physically interpretable sensitivity patterns from those that may be influenced by high-dimensional sampling effects. We will also explicitly contrast the HRU-scale results with the subbasin-scale analysis, which is less affected by dimensionality and therefore provides a more robust baseline for physical interpretation.

**Comment 5:** The use of NSE as the sole sensitivity target can be problematic, especially for rolling windows. NSE is highly sensitive to variance and peaks, can behave poorly for low flows, and is nonlinearly bounded; windowed NSE can become unstable for low-variance windows. You mention this as a limitation, but the paper's main results rely on it. You might want to include at least one complementary metric and a compare whether dominant parameters/hot moments persist across metrics. Otherwise, the "hot moments" identified (e.g., CN2 peaking during wet months) may reflect the flaws or the mathematical structure of NSE rather than the shift in the physical process.

**Reply:** We thank the reviewer for this insightful comment and agree that relying on NSE as the sole sensitivity target has limitations, particularly for rolling windows and low-variance periods. NSE can emphasize peak flows and may introduce metric-dependent artifacts in time-varying sensitivity analyses.

In the original manuscript, NSE was selected because it is the most widely used performance metric in SWAT-based studies and provides consistency with existing literature. Nevertheless, we acknowledge that conclusions based solely on NSE should be interpreted with caution.

Following the reviewer's suggestion, we will include at least one complementary performance metric (e.g., KGE) as a robustness check to examine whether the identified dominant parameters and hot moments persist across metrics. We will also temper the interpretation of the results to explicitly acknowledge the dependence of hot-moment identification on the chosen performance metric.

**Comment 6:** The manuscript states the model was "simulated at a monthly time step … consistent with available meteorological data," yet earlier you describe daily meteorological records and daily runoff data availability. Clarify if SWAT time step was daily or monthly or whether outputs were aggregated to monthly for NSE/SSA.

**Reply**: We apologize for the confusion caused by this wording. The meteorological inputs are available at a daily resolution, whereas the streamflow observations used for model evaluation are available at a monthly resolution. For consistency with the runoff data and to reduce computational cost, the SWAT simulations were performed at a monthly time step. We will correct this description in the revised manuscript and clarify the rationale for using a monthly simulation time step.

**Comment 7:** You highlighted the use of lagged Spearman's rank correlations between sensitivity and runoff is a highlight. However, the 1-month lag in 3-month windows needs clearer physical attribution. Is this a signature of soil moisture memory or a delay in the surrogate response?

**Reply**: We thank the reviewer for this insightful comment. We agree that the physical interpretation of the observed 1-month lag requires clarification. A plausible explanation is soil moisture memory, which can introduce delayed hydrologic responses and lead to temporal shifts in sensitivity patterns along the river network. At the same time, we acknowledge that the monthly simulation time step used in this study limits our ability to clearly distinguish physical memory effects from methodological or temporal aggregation effects.

In the revised manuscript, we will therefore avoid over-interpreting the lagged correlations and explicitly discuss alternative explanations, including soil moisture memory and the influence of temporal resolution. We will clarify that the lagged relationships are indicative rather than definitive and that higher-resolution simulations would be required to more rigorously attribute the underlying physical mechanisms.

**Comment 8:** It seems that CN2 is described as adjusted by a multiplicative factor in screening, and then later treated as independently distributed at subbasin and HRU scales. The replacement vs factor approach is not consistently explained. The read might want to know how "distributed parameters" are represented and thus what the Sobol indices refer to. You might want to describe how key parameters are perturbed at each scale (subbasin vs HRU), including how SWAT input files are edited and whether spatial structure is preserved or broken for the SA purpose. This would help reproducibility if other researchers are interested in this approach.

**Reply:** We thank the reviewer for pointing out this lack of clarity. The apparent inconsistency arises from insufficient explanation rather than from a change in parameter treatment. In all cases, CN2 is perturbed using a multiplicative factor; however, the level at which this factor is applied differs between analysis stages. During the screening stage, a single multiplicative factor is applied uniformly, whereas in the subbasin- and HRU-scale Sobol analyses, independent multiplicative factors are assigned to each spatial unit.

We agree that this distinction was not clearly described in the original manuscript. In the revised version, we will clarify how distributed parameters are represented at each scale, explicitly describe how SWAT input files are modified for the sensitivity analysis, and indicate whether spatial structure is preserved or intentionally relaxed. We will also add a schematic workflow to illustrate parameter perturbation across the screening and Sobol stages, thereby improving transparency and reproducibility.

**Minor Comments**

**Comment 1:** The largest errors occur during high-flow months, attributed to the "limited representation of extreme events in the training dataset". For a model intended to support "flood warnings," this is a significant deficiency.

**Reply:** We thank the reviewer for pointing this out. We agree that the larger errors during high-flow periods highlight an important limitation of the surrogate model, particularly for applications related to flood warning. In the revised manuscript, we will explicitly emphasize that the surrogate

model is not intended to replace process-based flood simulation, and that its reduced performance during extreme events reflects both the limited representation of high-flow conditions in the training data and the inherent difficulty of learning rare extremes.

**Comment 2:** There is a duplicated/incorrect subsection header ("2.5.2 Spatial Parameterization…" and 2.5.1 Spatial Parameterization for Distributed Parameters)

**Reply:** We apologize for this mistake. The subsection title of Section 2.5.2 will be corrected to "Construction of Deep-learning Surrogates".

**Comment 3:** Numerous typos/grammar issues (e.g., "trransform", "unfiorm", "predication error", etc.)

**Reply:** We apologize for these errors. The manuscript will be carefully proofread, and all typographical and grammatical issues will be corrected in the revised version.

**Comment 4:** Resampling land use to 3000 m and DEM to 150 m is a major preprocessing decision; provide quantitative justification that hydrologic response and HRU composition are not materially altered and implications.

**Reply:** We thank the reviewer for highlighting this important preprocessing choice. The resampling resolutions for the DEM and land use data were selected based on a series of quantitative sensitivity tests to ensure that the basin geometry, drainage structure, and HRU composition were not materially altered.

Specifically, we tested DEM resolutions ranging from 30 to 3000 m (30, 60, 90, 150, 300, 500, 750, 1000, 2000, and 3000 m) and examined their effects on watershed area, number of subbasins, river network length, and drainage density. The results indicate that for DEM resolutions up to 150 m, subbasin boundaries and drainage characteristics remain highly consistent with those derived from the original 30 m DEM, whereas coarser resolutions lead to clear reductions in basin area and subbasin number. Based on these results, a DEM resolution of 150 m was selected.

In addition, land-use data were resampled to multiple resolutions and combined with different DEM resolutions to assess their influence on HRU composition (Figure R2). This analysis shows that HRU numbers are primarily controlled by DEM resolution, while land-use resampling exerts a comparatively smaller influence under a fixed DEM resolution. The selected combination (DEM = 150 m, land use = 3000 m) therefore represents a compromise between preserving the essential hydrologic structure of the basin and maintaining computational feasibility for the large number of model evaluations required.

Detailed quantitative results and supporting figures will be provided in the Supporting Information, and the implications of these preprocessing choices will be briefly discussed in the revised manuscript.

[Figure]

**Figure R2.** Number of HRUs derived from DEM and land use data at different spatial resolutions

**Comment 5:** The "hierarchical calibration strategy" is discussed, but the study does not actually demonstrate calibration improvement; tone this down or add an illustrative experiment.

**Reply:** We thank the reviewer for pointing this out. In the revised manuscript, we will therefore tone down the related statements and clarify that the calibration strategy is discussed conceptually rather than empirically demonstrated.

**Recommendation**

The workflow is promising, but the paper needs stronger validation, clearer reproducibility, and more cautious interpretation.

**Reply:** We thank the reviewer again for the overall assessment and recommendation again.